# Intervention for Children with Obesity and Overweight and Motor Delays from Low-Income Families: Fostering Engagement, Motor Development, Self-Perceptions, and Playtime

**DOI:** 10.3390/ijerph19052545

**Published:** 2022-02-22

**Authors:** Adriana Berleze, Nadia Cristina Valentini

**Affiliations:** Universidade Federal do Rio Grande do Sul, Rua Felizardo 750, Porto Alegre 90690-200, RS, Brazil; pennanato@gmail.com

**Keywords:** child development, intervention, BMI, motor skills, perceived competence, obese children, overweight children

## Abstract

Obesity is increasing globally, affecting children’s health and development. This study examines the influence of a motor skill intervention on the daily routine, self-perceptions, body mass index, motor development, and engagement in physical education lessons of children with obesity and overweight with motor delays. Children were randomly assigned to intervention and control groups. The daily routine at home, self-perceptions, motor development, BMI, and engagement were assessed. Significant group by time interactions were found for play (*p* < 0.0001) and television (*p* < 0.0001) time, perceived social (*p* = 0.003) and motor (*p* < 0.0001) competence, global self-worth (*p* < 0.0001), BMI (*p* = 0.001), motor development (*p* < 0.0001), and engagement (*p* = 0.029). From pre-to-post intervention, children with obesity and overweight in the intervention group increased (1) playtime at home; (2) self-perceptions of social and motor competence and global worth; (4) engagement in the lessons, and improved scores, in motor skills; and (6) reduced BMI and screen time. The intervention promoted the health and improved the self-concept of children with obesity/ overweight.

## 1. Introduction

Obesity is increasing all over the world, affecting children’s health [1], motor development [2,3], and physical activity engagement [4]. Although the effects of the continuing increase in childhood obesity are not entirely understood, family and obesogenic social contexts [5], sedentary behaviors [6], and lack of physical activity [7] explain the noticeable escalation rate of obesity. Obesity also presents risks for development; children with obesity and overweight show motor delays [3,8,9].

The treatment of childhood obesity is a matter of prevention. Besides providing strategies to children to respond to obesogenic contexts [7], engaging children in motor activity is crucial—cardiorespiratory fitness and strength activities may even attenuate the genetic disposition to high BMI [10,11]. However, physical activity interventions have little effect on children with overweight and obesity [12]; the lack of evidence supports the need to develop more effective strategies. Regarding motor skills interventions, participating in a Mastery Motivational Climates (MMC) initiative promotes improvements in motor development, self-worth [13] and physical activity commitment and enjoyment [14] for obese and overweight children. 

Mastery Motivational Climate—a child-centered approach grounded on the achievement goal theory [15,16,17]—supports the conception that the nature of children’s experiences and how they interpret these experiences influences their motivation and achievement behavior. Central to the MMC is recognizing that efforts and outcomes are related— effort leads to personal progress and mastery of goals and skills. The comprehension of the effort–outcomes relationship leads children to build adaptative patterns of motivation, persisting in the tasks even when facing difficulties and failure, and consequently has a long-term effect on learning across the lifespan [15,16]. A considerable number of mastery climate studies have incorporated the TARGET framework (Tasks, Authority, Recognition, Grouping, Evaluation, Time) to implement specific strategies within this climate in the educational context to foster children motivation [18,19,20,21,22].

MMC is established when the teacher structures the classroom to convey goals, cues, strategies, and expectations that emphasize children’s autonomy in the learning process. In mastery climates, the child plays a cooperative role in establishing the rules; effort and hard work are privately recognized; challenging tasks and individualized evaluation are provided to promote the motivation to learn to move [19,20,21,22]. Although evidence of its effectiveness is provided for children with delays [19,20], the research is still limited for children with obesity and overweight [13,14]. Furthermore, children’s engagement behaviors during the mastery climate intervention sessions have not received research attention [22]. Insights about the child’s engagement may support the implementation of more effective teaching strategies. 

This randomized controlled trial examines the influence of an MMC motor skill intervention on the daily routine, self-perceptions, BMI, motor development, and engagement of children with obesity, overweight, or adequate weight but with motor delays. Considering that MMC provides similar opportunities for children with different levels of learning ability [19], we expected that children with obesity and overweight in the MMC group would demonstrate positive and similar patterns of improvement in motor development, self-perceptions, and engagement; and a reduction in BMI, waist circumference, and sedentary routine, from pre- to post-test, compared to their peers with adequate weight. We also expected that children in the MMC group would show more significant improvements in motor development, self-perceptions, engagement, and a more robust reduction in BMI, waist circumference, and sedentary routine, from pre- to post-test, compared to their peers in the control group.

## 2. Materials and Methods

### 2.1. Participants

The study’s sample estimation, using G-Power, was conducted considering the ANOVA repeated measure, with within and between interaction model as the primary analysis. The estimation was conducted considering 2-groups (Intervention and Control) × 2-nutritional status (Obesity/Overweight and Adequate Weight) × 2-time points (Pre- to Post-intervention), a power of 0.80, a significance level of 0.001, an effect size r = 0.20, and probable missing data of 30%. A sample size estimation indicated the need for approximately 100 children (50 for each group) for the RTC design. The study was announced to the local primary health care units. Children (6 and 7 years old) from low-income families (maximum of 2 minimum wages in Brazil; approximately US $450 monthly) were included based on the following criteria: being overweight or with obesity (O/O: BMI 85 ≤ percentile) [23] or adequate weight (AW) and with an indication of motor delays. The enrolment strategy targeted the specific quota of 1 child O/O: 1 child AW ratio, both with delays. Children (*n* = 120) referred to participate in the study were assessed by the Test of Gross Motor Development-2 (TGMD-2) [24]; 100 children from low-income families, 54 with O/O and 46 AW, met the inclusion criteria for motor delays (scored ≤ 35th TGMD-2 percentile); 20 children with adequate weight did not meet the criteria for motor delays. We randomly assigned children to the Mastery Climate Group (MCG: *n* = 50) or Control Group (CG: *n* = 50), stratified by weight status. The randomization was conducted by an independent researcher, not enrolled in the present study, using a free online software Randomizer (https://www.randomizer.org accessed on 10 December 2017), and was concealed from participants, families, and the two researchers responsible for the present study. The university ethical committee approved the study; the study was registered retrospectively in the national database for clinical trials (RBR-8t9bqkg). All parents signed the informed consent. Children verbally agree to participate. Participants were informed about their rights to decline participation or to withdraw from the study at any time.

### 2.2. Instruments and Procedures

#### 2.2.1. Children’s Routine at Home

A questionnaire on children’s daily routine was used [25]. The instrument has 12 questions, with multiple choice answers, organized in five dimensions related to (1) children’s transportation from home to school (i.e., car, bus, bicycle), (2) physical spaces for the child to play in during free time (i.e., parks, back yard, inside the home), (3) frequent play activities (i.e., board games, coloring, drawing, music) and chores at home (i.e., cleaning the room, help with siblings, helping take care of the house), (4) children’s interactions with other children (i.e., friends to play with from neighborhood, school), and (5) administration of the children’s time at home in different activities. We adapted the questionnaire for the last dimension and inserted a question about the time (minutes) children spend daily using a computer, watching television, and playing (i.e., fine and gross physical forms of play during free time). We assessed children’s routine at home for five consecutive days (not including the weekends) in the first and last week of the intervention; parents completed the questionnaire. The questionnaire has been used in previous research in Brazil [26], and, in the present sample, adequate internal consistency (α 0.78) and test–retest temporal stability (one week-interval; r = 0.83) were found.

#### 2.2.2. Self-Perception

The subscales, social acceptance, motor competence, and global self-worth (i.e., cchildren’s perception of how much they like themselves as a person) of the Pictorial Scale of Perceived Competence and Social Acceptance [27], validated for Brazilian children [28], were used to assess children’s self-perception. The scale has a structured pictorial Likert response format (1 lowest to 4 highest).

#### 2.2.3. Body Mass Index

Height was measured while the child stood straight with the assessor adjusting the horizontal lever using a portable stadiometer to the skull’s apex. Weight was measured using an electronic calibrated scale. BMI cutoff points were calculated according to WHO curves (percentiles: underweight <3; adequate weight 3≤ to <85; overweight 85≤ to <97; obesity ≥97) [23].

#### 2.2.4. Motor Skills

The TGMD-2 [24] validated for Brazilian children [29] was used to assess children’s fundamental motor skill performance individually. All tests were video recorded, and two independent raters, with extensive training, coded children’s performance; raw scores were used. High levels of inter-rater reliability were found for locomotor skill (LOC: Intraclass Correlation Coefficient—ICC = 0.93; range from 0.91 to 0.96) and Object Control skills (OC: ICC = 0.92, range from 0.89 to 0.94).

#### 2.2.5. Engagement in Intervention Lessons

The engagement in the sessions was assessed and coded using an observational procedure [30]. Several behavioral categories were coded: appropriate motor engagement with success (i.e., the child was successful in executing a skill or completing the task), appropriate engagement without success (i.e., the child makes mistakes during the process or product of the action), free play (i.e., the child engages in activities that were non-relevant to the tasks), changing tasks (i.e., the child identified the task but chose to practice another skill), distractions (i.e., the child enrolled in talk with others and stopped the practice), and conflicts (i.e., the child joined events that cause physical or verbal harm). Cameras were placed around the room; six sessions (first and last weeks) were recorded for further coding by two trained examiners. The coding started as the child began the practice in the stations, and the examiner observed the child for four minutes, categorizing the observed behaviors—the observations and scores restarted with another child every four minutes. The coders were trained with approximately 20–25 h of coding videos. After achieving 95% agreement during training, they started to code the children’s behaviors in the present study. A high level of concordance was obtained (ICC = 0.97).

The independent raters for motor skills performance and engagement behaviors during the lessons were blinded to each other’s assessment, the intervention period of assessment (pre-or post-tests), and children’s groups (MCG or CG); the raters did not participate in the intervention design, lesson planning, and implementation.

### 2.3. Design and Implementation 

#### 2.3.1. Mastery Climate Group

Children in the MCG participated in a 28-week-Mastery Motivational Climate Motor Skill Intervention (56 sessions/2 times per week/90 min each), emphasizing motor skills (41 sessions) and healthy eating habits orientation (15 sessions). Two intervention groups were formed with 25 children each, in order to have a manageable class; the two groups attended on Monday/Wednesday and Tuesday/Thursday. The intervention sessions for the MCG were planned consistently within the MMC strategies and the TARGET (Tasks, Authority, Recognition, Groups, Evaluation, Time) structure, a child-centered approach with high autonomy for children [19,20,21].

Regarding tasks, a variety of appropriate motor tasks was implemented in stations, challenge pathways, and games. Each station had several difficulty levels within the tasks and various equipment to adjust to a wide range of skill levels. Some sessions provided nutritional content games. The nutrition-based physical activity sessions combined movement practice with knowledge about healthy eating habits, groups of foods and the food pyramid, seasonal food, fresh food, daily portions of different groups of food, and healthy choices among various foods. The lessons were fun and conducted in stations, during games, and even in different veggie-fruits dance–sing–music activities at the end of the lessons. Parents participated in six sessions (session number 7, 14, 21, 28, 35, 42 for 90 min in each session), practicing motor skills and playing nutritional-based activity games with the children under the teacher’s instruction. All stations included nutritional orientation, such as (1) plate relay race—choose pictures of food, put in the bin and take to the next child to complete the race with a healthy plate; (2) design healthy food in balloons and beach balls and use to strike, kick or throw; (3) relays where children had to choose from different pictures of food and complete their plates with healthy choices; (4) stations where the child kicked a ball or threw a bean-bag, with colors representing the food pyramid, in the respective colors of a food-pyramid target; (4) locomotor pathway where the child takes a card with a food picture, gives the name of the. food and why it is good for your health and then takes a card with a motor challenge (i.e., jump, hop, run, slide) and completes the pathway with different challenges. All parents took part in the lessons in a minimum of four sessions; 22% of the parents were unable to attend the other two sessions; however, a relative and or an older sibling attended the session instead.

Regarding authority, children manage their time and choose from diverse levels of task difficulty within the stations. Children have an active role in establishing rules and individual short-term goals during the intervention lessons. They also created tasks to be included in the session with given equipment (i.e., balls, ropes, cones, bean bags, hoops, rackets, bats, mats, rings) and proposed different task difficulty levels (e.g., throw to different distances and target sizes, jump for height using different sizes of boxes, catch balls projected at different velocities).

Regarding recognition, parents were enrolled in the process of acknowledging children’s accomplishments. Parents received notes about the child’s progress and were encouraged to use this information to acknowledge their child’s efforts. Children received reinforcement and individual and group praise. Parents also participated in six lessons to acknowledge children’s motor progress and knowledge about nutrition and health.

Regarding groups, children had the opportunity to organize groups, choose peers, and practice in small heterogeneous groups. The groups were flexible during the stations’ practice, since children chose the stations and when to move. Children elected their practice groups. 

Regarding evaluation, an individual and group evaluations related to children’s participation and positive behavior were provided. To guide children in developing self-evaluation, we encouraged them to assess their attitudes toward learning and evaluate their motor skills during practice using verbal cues to describe the skills.

Regarding time, the number of stations allocated for each skill was based on the initial level of children’s motor skills. Although children had choices related to which station they would practice at, the flow of children away from completely occupied stations to reduce waiting time was conducted when necessary.

#### 2.3.2. Control Group

Children in the CG participated in a 28-week teacher-centered approach, a low-autonomy climate, and physical education sessions (56 sessions/2 times per week/90 min each). The lessons were planned to be aligned with the school curriculum, emphasizing the instruction and practice of fundamental motor skills with a specific focus on run, jump, hop, catch, and throw; a variety of games (i.e., tag, target, relay, and invasion), and recreational activities that mainly concern children’s choices of free play with and without equipment. The groups were also organized with 25 children and attended the lessons on alternate days. All the activities were appropriate for the child’s development. The prevailing, more traditional climate in sessions was also described, along with the TARGET structure. 

Regarding task dimension, the emphasis was on fundamental motor skills practiced in various small and large games, such as soccer and tag games. For the small games, a variety of equipment was used. Modeling and verbal instructions were provided along with the games, although the teacher constantly focused on acceptable behavior. Children had no choice of tasks. 

Regarding authority, direct instruction was implemented for motor skills content; the focus was on following teacher instruction and adequately completing the activities and tasks within a given time. During recreational games, children had the freedom to choose what they would like to play; only active games were allowed. The teacher established the protocol of acceptable behavior, and the rules were practiced along during the sessions. 

Regarding the recognition dimension, the teachers encouraged children’s efforts, achievements, and progress. During the games, the teacher verbally reinforced children’s accomplishments (i.e., score). 

Regarding groups, children were organized by the teacher in pairs, small groups, and large groups for most activities; children had the opportunity to choose teams in the large games. 

Evaluation strategies were mainly using positive feedback to motivate children during the lessons. During the activities, the teacher provided a group evaluation and general instructions. The teacher also provided individual evaluations regarding acceptable or inappropriate behavior during the lessons.

Regarding time, children completed the tasks to a specific schedule.

A doctoral candidate in human movement science, who was a physical education teacher with ten years of public-school experience, delivered the interventions. Children were assessed at pre-and post-test intervention by a trained professional with extensive experience in. assessment and with the help of two master students. Research design is presented in Figure 1.

### 2.4. Data Analysis

The frequencies for the daily routine were analyzed using Chi2 (between group comparisons) tests and Macnemar’s test (within group comparison). A 2 (groups: MCG and CG) × 2 (nutritional status: O/O and AW) × 2 (time: Pre- to Post-intervention) ANOVA with a repeated measure on the last factor was conducted to examine the intervention’s effect on daily time at play and screen (computer and television), self-perceptions, BMI, waist circumference, LOC skills, and OC skills. A 2 × 2 ANOVA with a repeated measure for the group factor was used to analyze the children’s engagement in lessons, and the pre-to-post intervention effects were analyzed. Partial η^2^ (eta squared) was used as the index of effect size for the ANOVA (η^2^ small = 0.01, moderate = 0.06, large = 0.14). Post hoc tests were reported only for significant interactions related to the study hypotheses (children with obesity/overweight: MCG × CG; children with adequate weight: MCG × CG), using Cohen’s D as the index of effect size (d: very small = 0.01, small = 0.20, medium = 0.50, large = 0.80, very large = 1.20). In addition, change scores (Delta: Δ scores) were also provided for each group in both conditions; comparisons were conducted for delta scores with Cohen effect size reported

## 3. Results

### 3.1. Children’s Daily Routine 

The results showed that most children at the pre-and post-test walked or rode bicycles to school, played in the backyard and nearby parks, drew, read, and watched TV at home. Half of the children did not have a computer. The MCG children with O/O increased the frequency they played (running, ball, riding a bike, and jump rope), whereas children with AW increased running and jump rope frequencies. The only change for the CG was the increase in running frequencies for children with O/O. Table 1 presents the children’s daily routine by group.

The ANOVA showed a significant interaction for playtime, F (3, 70) = 11.80, *p* < 0.0001, η2 = 0.33, with a large effect size. Post-hoc tests showed that playtime was similar at the pre-test (*p* = 0.314) and different at the post-test (*p* < 0.0001); MCG-O/O spent more time playing at the post-test than CG-O/O, with very large effect size (*p* < 0.0001; d =1.67). MCG-O/O (*p* < 0.0001), and MCG-AW (*p* = 0.016) increased playtime from pre- to post-tests. 

The ANOVA showed a significant interaction for television time, F (3, 70) = 13.97, *p* < 0.0001, η2 = 0.37, with a large effect size. Post hoc tests showed that TV time was similar at the pre-test (*p* = 0.367) and different at the post-test (*p* < 0.0001); MCG-O/O (*p* = 0.004; d = 1.06) and MCG-AW (*p* = 0.003; d = 1.55) spent less time watching TV than the CG, with large effect sizes. The MCG-O/O (*p* < 0.0001) and MCG-AW (*p* = 0.006) decreased time watching TV from pre- to post-test. Computer time interaction was non-significant (F (3, 68) = 2.01, *p* = 0.121, η2 = 0.08). Figure 2 presents the scores for play and screen time by group. 

Please refer to Appendix A for the daily play, computer, and TV time descriptive scores, independent and dependent t-tests, and Delta comparisons for MCG and CG.

### 3.2. Self-Perceptions 

The ANOVA showed a significant interaction for perceived social acceptance, F (3, 74) = 18.72, *p* = 0.003, η2 = 0.17, with a large effect size. Post hoc tests showed that groups were similar at the pre- (*p* = 0.354) and post- (*p* = 0.176). Scores increased from pre- to post-tests for MCG-O/O (*p* < 0.000) and MCG-AW (*p* = 0.001). 

The ANOVA showed a significant interaction for perceived motor competence, F (3, 74) = 13.65, *p* < 0.0001, η2 = 0.36, with large effect size. The post hoc tests showed that groups were similar at pre-test (*p* = 0.178) and different at post-test (*p* = 0.013); MCG-O/O (*p* = 0.046; d = 0.67), and MCG-AW (*p* = 0.010; d = 1.21) demonstrated higher scores than CGs in the post-tests, with medium and large effect size. Scores increased from pre- to post-test for MCG-O/O (*p* < 0.0001) and MCG-AW (*p* < 0.0001). 

The results from the ANOVA showed a significant interaction for global self-worth., F (3, 74) = 11.70, *p* < 0.0001, η2 = 0.32, with a large effect size. Post hoc tests showed that groups were similar at the pre-test (*p* = 0.152) and different at the post-test (*p* = 0.039); MCG-AW showed higher scores than the CG-AW (*p* = 0.010; d = 1.21). Scores increased from pre- to post-test for the MCG-O/O (*p* < 0.0001) and MCG-AW (*p* < 0.0001). Figure 3 presents the scores for self-perceptions by group.

Please refer to Appendix A for perceived social acceptance, motor competence and global self-worth descriptive scores, independent and dependent t-tests, and Delta comparisons for MCG and CG.

### 3.3. Body Mass Index and Waist Circumference

Prevalence of obesity/overweight was similar at the pre-test; at the post-test, the prevalence of obesity declined by 18.50% for the MCG and 10.00% for CG. The 2 × 2 × 2 ANOVA showed a significant interaction for BMI, F (1, 74) = 7.12, *p* = 0.001, η2 = 0.22, with a large effect size. Post hoc tests showed that BMI were similar at the pre- and post-tests (*p* = 1.00); and decreased from pre- to post-test for MCG-O/O (*p* < 0.0001) and CG-O/O (*p* = 0.032). For waist circumference, the interaction was non-significant, F (3, 74) = 0.39, *p* = 0.772, η2 = 0.01; however, a significant pre-to-post intervention reduction was observed for children with O/O in the MCG. Figure 4 presents the BMI (4a) and waist circumference (4b) scores by group.

Please refer to Appendix A BMI and waist circumference descriptive scores, independent and dependent t-tests, and Delta comparisons for MCG and CG.

### 3.4. Motor Skills

The ANOVA showed a significant interaction for LOC skills, F (3, 74) = 17.80, *p* < 0.0001, η2 = 0.42, with large effect size. Post hoc tests showed that scores were different at the pre- (*p* = 0.001) and post- (*p* < 0.0001) tests; MCG-O/O (Pre-test: *p* = 0.042, d = 0.84; Post-test: *p* < 0.0001, d = 2.71) and MCG-AW (Pre-test: *p* = 0.007, d = 1.32; Post-test: *p* < 0.0001, d = 3.55) showed higher scores than CGs, with large effect size. All groups increased scores from pre- to post-test (MCG-O/O *p* < 0.0001; MCG-AW *p* < 0.0001; CG-O/O *p* = 0.019; CG-AW *p* = 0.003).

The ANOVA showed a significant interaction for OC skills, F (3, 74) = 30.04, *p* < 0.0001, η2 = 0.55, with large effect size. Post hoc tests showed that scores were similar at the pre-test (*p* = 0.119) and different at the post-test (*p* < 0.0001); MCG-O/O (*p* < 0.0001); MCG-AW (*p* < 0.0001) exhibited higher scores than the CGs at post-test. All groups increased scores from pre- to post-test (MCG-O/O *p* < 0.0001; MCG-AW *p* < 0.0001; CG-O/O *p* = 0.013; CG-AW *p* = 0.015). Figure 5 presents the LOC (5a) and OC (5b) skills scores by group.

Please refer to Appendix A for the TGMD-2 descriptive scores, independent and dependent t-tests, and Delta comparisons for MCG and CG.

### 3.5. Engagement in the Intervention Lessons

The time effects for appropriate motor engagement with-success (F (1, 37) = 292.72, *p* < 0.0001, η2 = 0.89), free-play (F(1, 37) = 10.65, *p* = 0.003, η2 = 0.22), changing-tasks (F (1, 37) = 23.68, *p* < 0.0001, η2 = 0.40), conflicts (F(1, 37) = 10.80, *p* = 0.002, η2 = 0.23), and distraction (F (1, 37) = 5.18, *p* = 0.029, η2 = 0.13) were significant, with large effect sizes. The post hoc tests showed from pre- to post-test increases in engagement with success (MCG-O/O: *p* < 0.0001; MCG-AW: *p* < 0.0001); and decreases in free play (MCG-O/O: *p* = 0.017; MCG-AW: *p* = 0.046) changing tasks (MCG-O/O: *p* < 0.0001; MCG-AW: *p* < 0.010), conflicts (MCG-AW: *p* = 0.026), and distraction (MCG-O/O: *p* < 0.0001; MCG-AW: *p* = 0.003). The time effect for engagement without-success was non-significant (F (1, 37) = 1.68, *p* = 0.203, η2 = 0.04). Figure 6 present the engagement scores for the MCG. 

Please refer to Appendix A for Motor Engagement within the context, with descriptive scores, independent and dependent t-tests, and Delta comparisons for MCG.

## 4. Discussion

### 4.1. Children’s Daily Routine 

Children spend time inside the home drawing, reading books, and watching TV. The most frequent outside activity was walking or riding bikes to school since parents had no car; children played in the backyard and nearby small parks. Visits to parks were not frequent due to lack of security and were allowed only under adult supervision; a similar trend was reported previously for Brazilian children [31,32]. All children in the intervention group increased the frequencies of playing outside, running, and riding bikes; a plausible explanation was that the intervention motivated and gave children resources to play at home.

The intervention effectively reduced the daily time using computers and watching TV and increased playtime. The MCG-O/O increased approximately 40 min in-play games and decreased by nearly 50 min the combined time using a computer and watching TV; we found a similar trend for the MCG-AW. Since a higher percentage of body fat is associated with time watching TV [33], the changes observed seem to promise to improve children’s quality of life.

Particularly encouraging was the reduction of media time. Since none of the children had cellular phones, and only half had a computer at home, screen time was related to TV. At the pre-test, children devoted nearly three hours to watching TV every day, one hour above the recommendation for this age group. At post-test, TV time was around two hours, which was adequate for this age [34]. It is essential to highlight that reducing obesity risks by parental behavioral modification has shown positive effects on decreasing children’s TV time [35,36]. Here we added a new piece of evidence showing that providing children with resources for playing (skills and games knowledge) helps them trade off the time in front of a TV for more active behavior. 

### 4.2. Self-Perceptions 

Up to now, there was little evidence for the impact of the intervention on children’s perceptions of social acceptance [37], and none addressed these perceptions in O/O children, stressing the originality of our study. These children often feel less acknowledged by their peers and excluded [38]. The MMC effectively helped children with obesity and overweight to feel acceptable, preventing the risks of isolation. There were no dropouts for children in the MCG with O/O; these children increased engagement and playtime. The intervention’s adherence was fostered by strengthening children’s perceptions combined with attractive tasks, a strategy previously reported as efficient [39].

The increases in social acceptance perceptions are comparable to studies conducted with vulnerable children with motor delays enrolled in sport intervention [33,35]. Contradicting our findings, two early intervention studies for children with motor delays reported no changes in social acceptance [40,41]. It is essential to notice that these studies implemented a directive approach and shorter intervention periods, whereas we applied a high autonomy climate [19,20,42] for a more extended period. The strategies within the MMC (e.g., choose peers, practice with children with various ability levels) played a role in the increase in children’s perceptions of social acceptance. 

Regarding perceived motor competence, interventions with multi-modal physical activity [9,38] and community-based physical and motor activity [13] approaches reported improvements in motor perceptions in obese and overweight children, similar to our results. Our results are comparable with research that implemented MMC interventions and positively changed children’s perceived motor competence for at-risk [43] and with motor delays [19,20,37].

Nonetheless, few studies have examined the effect of motor interventions on self-worth. Previously, a multidisciplinary intervention program with overweight and obese children [38], and two MMC studies implementing a sport [37] and school based [39] interventions, reported improvement in global self-worth, like the gains observed for all children in the MCG. Here, we extended the previous evidence and supported MMC’s effect on social acceptance and global self-worth for children with O/O and AW.

### 4.3. Body Mass Index and Waist Circumference

All children with O/O reduced BMI; MCG-O/O showed more dramatic decreases than the CG-O/O. Reductions in waist circumference were observed only for the MCG-O/O. Previous interventions using nutrition and physical activity strategies for similar age groups showed decreased BMI and waist circumference [8,39,44]. Decreases in BMI are a promising result, considering the global scenery of increases in weight in children. 

Changes in BMI are challenging due to several environmental factors (e.g., unsafe parks, inactivity, restricted access to healthy food, large community criminality rates) that children and families have no control over [5]. Likewise, family habits of consuming sugar-added food play an undesirable role in increasing childhood obesity [45]. Since young children are more likely to have no control over these specific matters, the parents’ educational program which was implemented may have contributed to the decreases in BMI and waist circumference in a noticeable short time in the present study. However, none of our measures included parental behavior changes; therefore, this influence’s extension on our results was not objectively estimated.

### 4.4. Motor Skills

We found dramatic changes in motor scores for children in the MCG. The MCG-O/O results were comparable to those reported in a physical activity program for Australian [13], and Belgian [8] children, and in two motor skills programs for Italian [9] and Australian [46] children. Evidence of an MMC intervention in improving motor development for obese and overweight children was reported in Brazil [47], as in our findings. The motor development improvements in MCG-AW are supported by previous interventions designed to promote children’s physical activity [44,46], motor [19,20,42], and sports skills [48,49].

All children were from low-income families with no means of providing admissions to sports programs; children’s only opportunity to exercise was the intervention program. Furthermore, healthy food is more expensive in Brazil [45], and the families could probably not make drastic modifications in eating habits. The intervention provided an opportunity to improve skills, increasing the likelihood that these children became more active [47].

### 4.5. Engagement in the Intervention Lessons

The results showed that appropriate motor engagement with success for children within the mastery climate increased in a similar pattern. A previous study reported that obese and overweight children’s engagement, measured with accelerometers, was strengthened by MMC strategies [14]. These results are, to some extent, comparable with ours. Participating in programs that encourage decision-making, choices, challenges, and reinforce competence is critical to promoting engagement, regardless of whether the measure investigated is objective (i.e., by accelerometers) or qualitative, such as changes in behavior.

All MMC children engaged very little in activities outside stations or in free play, and changing tasks decreased along with the intervention, similar to a previous study with American children [23]. Besides, children with obesity/overweight demonstrated initially higher engagement in disturbances than children with adequate weight. Over the intervention period, this behavior decreased. These results are the consequence of children following the cooperative protocol established and having attractive stations to practice in. The stations were diversified with equipment that children usually do not have at home or school. The setting was a powerful attractor of children’s attention towards learning [22,50].

Children increased their engagement regardless of their weight. The time the children devoted when enrolled appropriately in the tasks, measured by frequency, or the emotional quality of involvement during the tasks, measured by the successful attempts to master a skill, increased for all children. Task-oriented behavior was related to the climate; previous studies supported more motivated behaviors for children in MMC interventions [22]; however, caution is recommended in interpreting these results since we lack the assessment of the control group for this outcome. Our results support the argument that intervention offers an effective means to improve motor development in childhood that may break through the circle of physical inactivity–motor deficits–frustration–increasing inactivity, and weight gain [14].

### 4.6. Study Limitations and Recommendations for Future Research

The present study has several limitations. First, we did not have an objective measure of children’s activities at home (i.e., pedometers or accelerometers). We used parental self-report measure; although this has advantages such as low-economic cost and provides parents with a quick view of the child’s time management at home, it also has some shortcomings. Specifically, it does not provide accurate estimates of the absolute amount of time in sedentary and physical activity; the entire amount of physical activity at home would be better estimated with an objective measurement; future studies may consider this.

Second, although all relevant for better understanding of children’s development, our outcome measures were only assessed immediately after the intervention. Consequently, the long-term effect of this intervention is yet unknown. Considering that all children were from low-income families and had restricted opportunities and resources to attend physical activity and youth sports programs, which are mainly private in Brazil, it is critical to understand whether this intervention will lead to long-term changes. Especially, children with obesity and overweight may require ongoing intervention to sustain decreases in BMI and screen time, increased playtime, engagement, self-perceptions, and motor skills.

Third, we did not control the family diet and parental changes in dietary behavior. Due to the families’ low socioeconomic status, we thought that any intention to control this factor could lead to family awkwardness and resistance to participating in the study. However, we recommend that further studies consider assessing these outcomes since their influence may go unnoticed.

Fourth, we did not measure children’s engagement in the control group lessons. Our first choice during the RTC design was to understand better the children’s engagement in mastery climate intervention, considering the lack of studies addressing this issue; more traditional teaching approaches, similar to those conducted in the control group, have more robust studies regarding the following engagement. We strongly recommend that further research measure this outcome in the control group.

### 4.7. Practical Implication 

Interventions are crucial for motor development and help to prevent childhood obesity. The efficacy of our results indicates that MMC intervention designed to improve motor and social parameters for children with obesity and overweight is feasible with a low-cost intervention that requires mainly the training of professionals to incorporate motivation strategies in the teaching process and to support child autonomy. Although our program was effective, only a small group of children benefited. Large-scale government interventions are necessary to enable primary strategies to prevent obesity; health adversity will not be reversed without governmental action. Government effort toward promoting research and treatment [1] is critical to improving children’s health. Prevention is an unconditional strategy to diminish the exposure to obesogenic risky environments and promote understanding about how these risks interact [5]. We provided evidence that a preventive intervention with the appropriate climate may control some risks (BMI, motor delays, poor engagement, low self-perception). Actions from individuals and organizations are necessary to improve children’s quality of life [5]. Given the pervasive and high prevalence of obesity in school-aged children, it is crucial to consider all scenarios to reduce childhood obesity; our contribution was to implement and assess several outcomes for children participating in an MMC intervention. 

## 5. Conclusions

Our primary contribution was to implement an intervention that effectively improved perceptions of social acceptance, motor development, global self-worth, motor development, appropriate engagement in the sessions, and reduced BMI and waist circumference. Children increased time playing and decreased the time watching TV at home. The changes were substantial for children with obesity, overweight, and adequate weight. The result may help discontinue the common perception that obese and overweight children are unmotivated and sometimes lethargic; they may need a suitable climate to feel motivated to engage. A limited number of interventions have been designed to address these concerns, and we have successfully demonstrated positive benefits for obese and overweight children.

## Figures and Tables

**Figure 1 ijerph-19-02545-f001:**
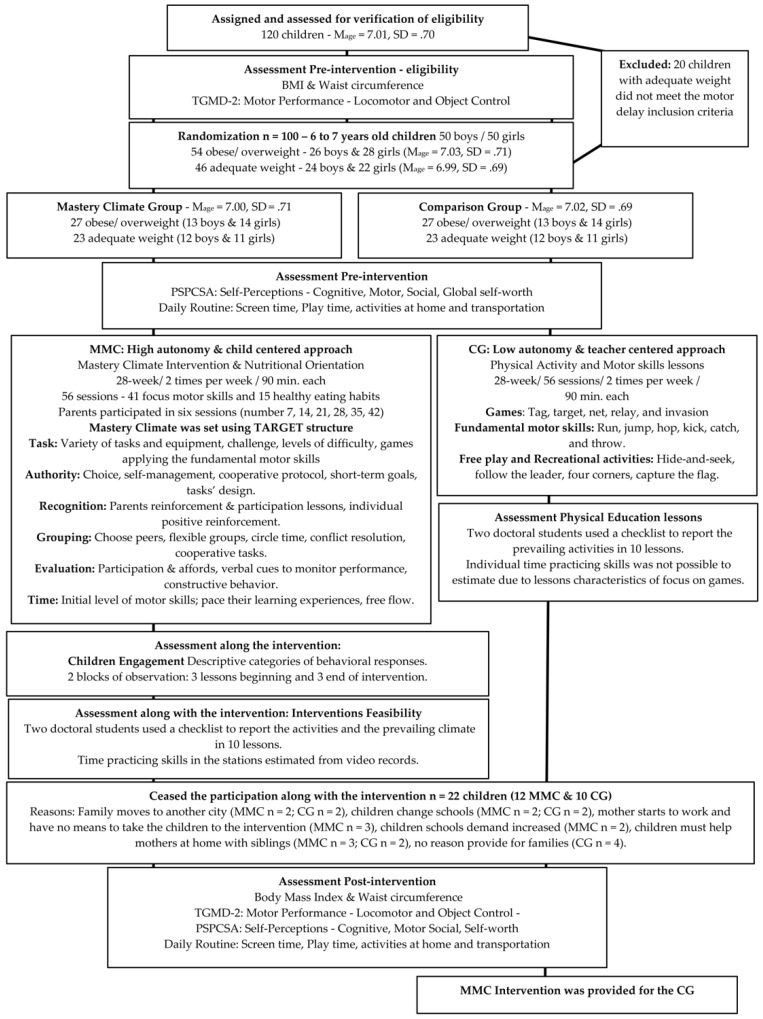
Research Design: Groups, assessments, and intervention.

**Figure 2 ijerph-19-02545-f002:**
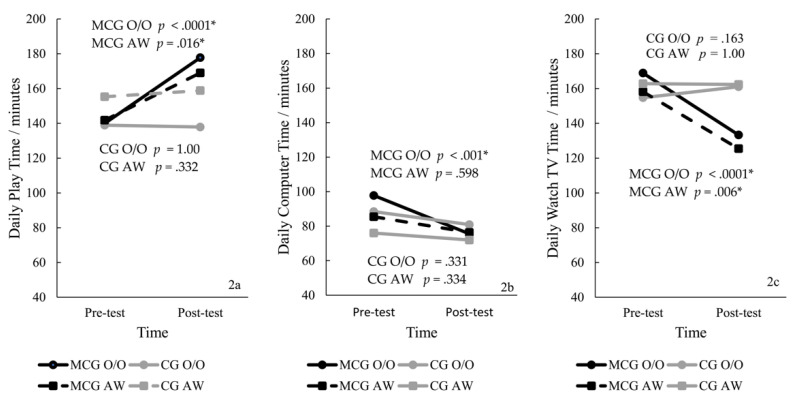
Daily play time (**2a**), computer time (**2b**) and watch TV time (**2c**) by groups (* significant results).

**Figure 3 ijerph-19-02545-f003:**
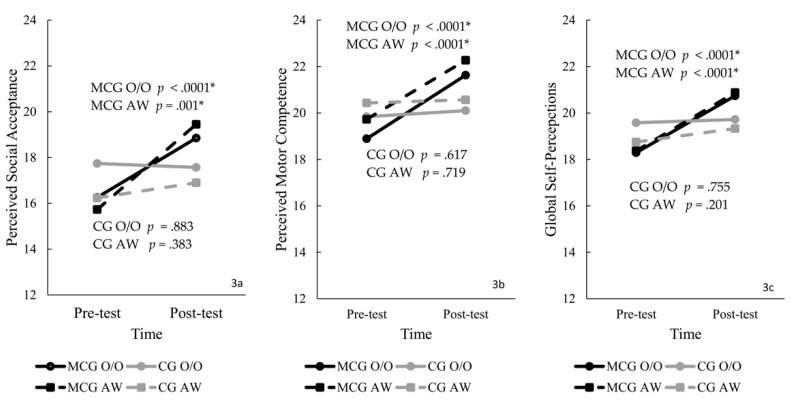
Perceived social acceptance (**3a**), perceived motor skill competence (**3b**) and perceived global self-perceptions (**3c**) by group (* significant results).

**Figure 4 ijerph-19-02545-f004:**
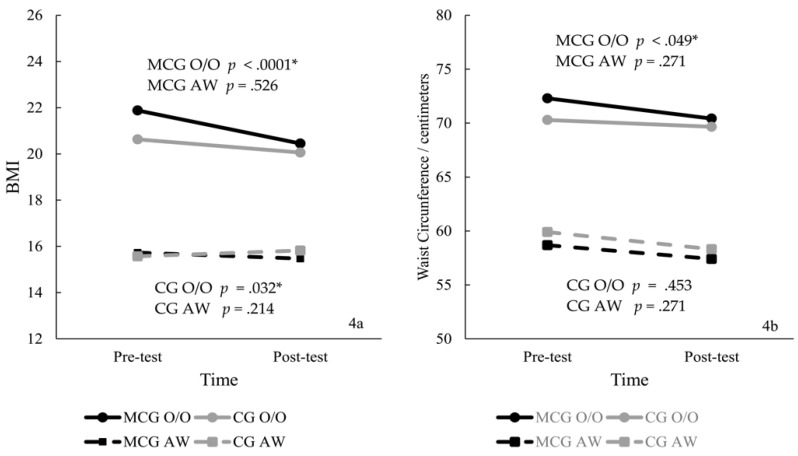
BMI (**4a**) and waist circumference (**4b**) by group (* significant results).

**Figure 5 ijerph-19-02545-f005:**
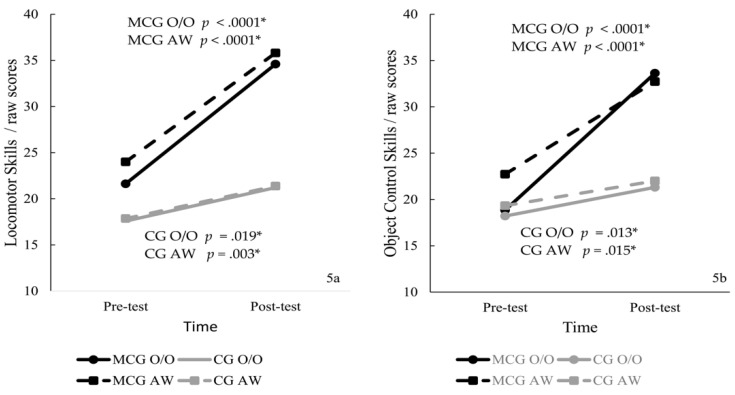
Locomotor (**5a**) and object control (**5b**) skills by group (* significant results).

**Figure 6 ijerph-19-02545-f006:**
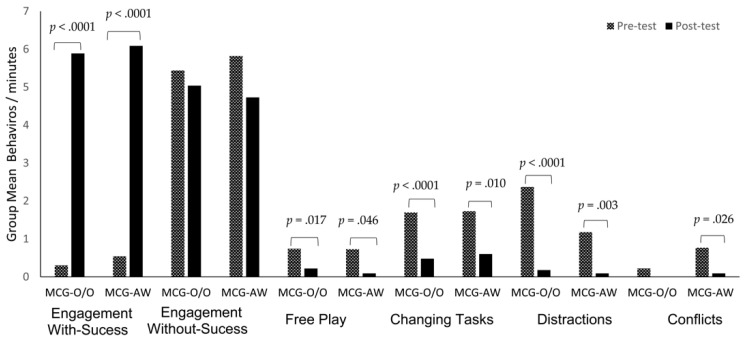
Children’s motor engagement behaviors in the Mastery Climate sessions by groups.

**Table 1 ijerph-19-02545-t001:** Children with obesity/overweight and adequate weight activities at home: MCG and CG.

		Children with Obesity/Overweight N (%)				Children with Adequate Weight N (%)			
Children Activities at Home		MCG		CG		MCG		CG	
		Pre	Post	Pre	Post	Pre	Post	Pre	Post
Mobility to school	Bus	14 (51.9)	12 (44.4)	4 (21.1)	4 (21.1)	3 (27.3)	2 (18.2)	4 (23.5)	4 (23.5)
	Walking/biking	13 (48.1)	15 (55.6)	15 (78.9)	15 (78.9)	8 (72.7)	9 (81.8)	13 (76.5)	13 (76.5)
Space to play	Park/Backyard	19 (70.4)	19 (70.4)	15 (78.9)	15 (78.9)	7 (63.6)	7 (63.6)	15 (88.2)	15 (88.2)
	Inside home	8 (29.6)	8 (29.6)	4 (21.1)	4 (21.1)	4 (36.4)	4 (36.4)	2 (11.8)	2 (11.8)
House’ chores	Usually	10 (37)	15 (55.6)	10 (52.6)	13 (68.4)	6 (54.5)	8 (72.7)	10 (58.8)	10 (58.8)
	Never	17 (63.0)	12 (44.4)	9 (47.4)	6 (31.6)	5 (45.5)	3 (27.3)	7 (41.2)	7 (41.2)
Run	2/3 times/week	14 (52.9)	25 (92.6) *	4 (21.1)	9 (47.4) *	5 (45.5)	9 (81.8) *	8 (47.1)	10 (58.8)
	None	13 (48.1)	2 (7.4)	15 (78.9)	10 (52.6)	6 (54.5)	2 (18.2)	9 (52.9)	7 (41.2)
Play ball	2/3 times/week	17 (63)	26 (96.3) *	14 (73.7)	17 (89.5)	9 (81.8)	11 (100)	16 (94.1)	16 (94.1)
	None	10 (37.0)	1 (3.7)	5 (26.3)	2 (10.5)	2 (18.2)	0	1 (5.9)	1 (5.9)
Dance and Circle Sing Games	2/3 times/week	4 (14.8)	6 (22.2)	6 (31.6)	6 (31.6)	5 (45.5)	6 (54.5)	8 (47.1)	9 (52.9)
	None	23 (85.2)	21 (77.8)	13 (68.4)	13 (68.4)	6 (54.5)	5 (45.5)	9 (52.9)	8 (47.10
Jump rope	2/3 times/week	9 (33.3)	24 (88.9) *	5 (26.3)	5 (26.3)	3 (27.3)	8 (72.8) *	3 (17.6)	4 (23.5)
	None	18 (66.7)	3 (11.1)	14 (73.7)	14 (73.7)	8 (72.7)	3 (27.5)	14 (82.4)	13 (76.5)
Ride Bike	2/3 times/week	20 (74.1)	26 (96.3) *	12 (63.2)	13 (68.4)	8 (72.7)	8 (72.8)	13 (76.5)	13 (76.5)
	None	7 (25.9)	1 (3.7)	7 (36.8)	6 (31.6)	3 (27.3)	3 (27.3)	4 (23.5)	4 (23.5)

Note: 4 families failed to return the questionnaire: valid percentage reported; *p* ≤ 0.05: Between group comparisons (MCG vs CG) were conducted using Chi^2^ tests; Within group comparisons were conducted using Macnemar tests *.

## Data Availability

The data presented in this study are available on request from the corresponding author. The data are not publicly available due to the lack of parental agreement.

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
