# Peer review of "Intervention for Children with Obesity and Overweight and Motor Delays from Low-Income Families: Fostering Engagement, Motor Development, Self-Perceptions, and Playtime"

_ijerph, 2022, doi:10.3390/ijerph19052545_

Round 1

Reviewer 1 Report

Dear authors, thank you for the opportunity to review your study on an intervention aimed to improve motor development, engagement, self-perception and playtime in children. Please see below for my comments and suggestions in order to make revisions to your manuscript.

  1. General comment: please consider completing the CONSORT checklist and TIDieR checklist to make sure that you have included all essential elements in your report. Many times, trials are designed with high quality but the reporting of the trial is inadequate.
  2. Abstract: it is not clear in the abstract what is the primary outcome or timepoint.
  3. Page 1 lines 28-30: the effects of obesity on motor development and physical activity were already mentioned in the first sentence, so this is repetitive. Please remove or combine references into the first sentence.
  4. Page 1 lines 31-32: there are many aspects to prevention, and education is only one of them. I suggest removing "and prevention is an educational subject".
  5. Page 2 lines 64-70: a large part of the instructions to authors have been inadvertently included in the manuscript (begins with "The Materials and Methods should...". This needs to be removed.
  6. Page 2 line 72: Please indicate which outcome measure was used for the sample size calculation
  7. Page 2 line 73: is this effect size Cohen's d? 0.6 is a pretty large effect for a therapy intervention. Do you have evidence from previous trials using the MMC approach that an effect size of 0.6 would be expected? Ordinarily, effect size estimates would be based on pilot studies in similar populations if available.
  8. Page 2 line 76: what was the criteria for being considered low income?
  9. Page 2 line 80: did you have a recruitment strategy or enrolment strategy to fill a particular quota of O/O vs. AW (i.e. aiming for a 1:1 ratio)? Explain how you managed to obtain the ratio, and if your recruitment strategy was random and not targeted, whether the proportion of children in your study that are O/O:AW is the same as in the community (i.e. is this a representative sample of low income families?).
  10. Page 2 lines 82-83: please provide more information about the randomization process, including who performed the randomization, how it was completed (e.g. sequential opaque envelopes/central telephone service/REDCap etc.), and whether it was concealed from participants and trial personnel until recruitment was confirmed. Please also indicate whether any blocking was used, and if so, what were the block sizes?
  11. Page 2 line 85: The trial was clearly registered retrospectively, as the date of registration was 10 January 2022. Please state that the trial was registered retrospectively to avoid doubt for the reader.
  12. Page 2 line 90: please provide information about the reliability and validity of the questionnaire for the children's daily routine. Since the reference is not an English language resource many readers will not be able to access it to obtain this information themselves. Furthermore, please consider providing information about how many questions it has and how it is scored.
  13. Page 3 2.3 Motor Skill Intervention Design and Implementation: A great amount of detail is provided for the MMC group. Please write another section providing the same level of detail about the control group. I recognize that there is some information provided in figure 1 however it is important for consistency to write the information in text also.
  14. Page 3 line 132: remove 'were' in "Children were participated in..."
  15. Page 3 line 133-134: please explain how the healthy eating habits sessions were structured in as much detail as possible (contents, framework etc.) and also explain how the healthy eating habits sessions were positioned in the 28 weeks (e.g., all motor skills sessions weeks 1-21 then all healthy habits sessions weeks 21-28 or interspersed?).
    If it is your intention to report the healthy eating habits results/intervention in another manuscript, please state your intention for this paper that you will focus only on the motor skills portion of the intervention.
  16. Page 5 Figure 1: Clarify at which point the assessment pre-intervention took place: before or after randomization? In the figure it appears that it happened after however I understand that TGMD-2 was applied first before randomization to eliminate participants who did not meet the motor delay criteria.
  17. Page 5 Figure 1: Break down the number of participants lost to follow up per treatment group rather than as a whole
  18. Page 5 Data Analysis: since your trial was registered retrospectively, readers will not be able to know whether your analyses were prospectively designed. Consider attaching a statistical analysis plan (dated) as a supplementary file to increase the quality of your reporting. You would have had to submit this as part of your ethically approved protocol.
  19. Page 5 Results: for transparency please provide the raw means and standard deviation scores and mean differences pre- and post-intervention for ALL outcomes that were analysed using ANOVA, even if it has to be in a supplementary data file. You have shown the graphs however these are difficult to extract data from, and it is likely that other researchers will seek this data for purposes of meta-analysis. Furthermore it is important to enable your statistical analyses to be verified. You may also consider publishing a copy of your dataset in a data repository which would serve the same purpose.
  20. Page 5 line 186: please use a consistent acronym throughout the whole paper for the intervention group, ideally MMC so that it does not also contain CG and be confused with the control group.
  21. Table 1: I am not sure why there are four columns. Is this because it is broken up into O/O and AW children? If so, there is a missing row at the top of the table that shows which columns are O/O and which columns are AW.
  22. Table 1 legend: should it read p<0.05 not p<0.5?
  23. Figures: This may have been done by the typesetter, but the graphs in figures are extremely small and are impossible to read. If it is within your control, please provide graphs that are as large as possible and in a high quality image format.
  24. Page 11: Consider adding a separate section where you detail some limitations of the study, for example that: 1. physical activity behaviours were measured subjectively using a self-report measure which may introduce bias (future studies should consider objective measurement with accelerometers), 2. outcomes were measured immediately post-intervention, so the longevity of changes in behaviour are not known. Given the environment (low-socioeconomic, low-resource) and because children have limited formal physical activity opportunities, it is critical to understand whether this intervention leads to a long-term change or whether ongoing intervention is required to sustain decreases in BMI, increased PA etc.
  25. Page 12 line 404: please expand the acronyms for the trial funders if relevant

Author Response

Thank you for the. nice and constructive comments, we addressed all the suggestions.

  1. General comment: please consider completing the CONSORT checklist and TIDieR checklist to make sure that you have included all essential elements in your report. Many times, trials are designed with high quality but the reporting of the trial is inadequate. Response: We agree; thanks for the suggestion. We use the TIDieR checklist to complement information regarding the intervention; several detailed information is provided now
  2. Abstract: it is not clear in the abstract what is the primary outcome or timepoint.Response: We clarify the primary outcomes and timepoints (lines 16 to 19).
  3. Page 1 lines 28-30: the effects of obesity on motor development and physical activity were already mentioned in the first sentence, so this is repetitive. Please remove or combine references into the first sentence. Response: We removed the repetitive sentence. We included reference 10 on line 34.
  4. Page 1 lines 31-32: there are many aspects to prevention, and education is only one of them. I suggest removing "and prevention is an educational subject". Response: We agree; it was deleted.
  5. Page 2 lines 64-70: a large part of the instructions to authors have been inadvertently included in the manuscript (begins with "The Materials and Methods should...". This needs to be removed. Response: We are sorry for the mistake. We deleted the instruction material. 
  6. Page 2 line 72: Please indicate which outcome measure was used for the sample size calculation. Response: Sorry for the lack of clarity. We rewrite the sentences for sample size estimation and reported it considering the statistical analysis adopted (ANOVA with repeated measure – pre-to-post intervention), the power level adopted (.80), the significance level of (.001), an effect size (eta squared .20), and an estimation of probable missing data of 30%.
  7. Page 2 line 73: is this effect size Cohen's d? 0.6 is a pretty large effect for a therapy intervention. Do you have evidence from previous trials using the MMC approach that an effect size of 0.6 would be expected? Ordinarily, effect size estimates would be based on pilot studies in similar populations if available. Response: No, we used eta squared as the effect size index in estimating the sample size, and we are sorry that we had a typo mistake (it should be .20 instead of .60). We rewrite the sample size estimation to clarify the procedures.
  8. Page 2 line 76: what was the criteria for being considered low income?Response: The criteria for the low-income class were included 
  9. Page 2 line 80: did you have a recruitment strategy or enrolment strategy to fill a particular quota of O/O vs. AW (i.e. aiming for a 1:1 ratio)? Explain how you managed to obtain the ratio, and if your recruitment strategy was random and not targeted, whether the proportion of children in your study that are O/O:AW is the same as in the community (i.e. is this a representative sample of low income families?). Response: we target a 1 O/O: 1 AW ratio. We inserted the information (page xx line xx)
  10. Page 2 lines 82-83: please provide more information about the randomization process, including who performed the randomization, how it was completed (e.g. sequential opaque envelopes/central telephone service/REDCap etc.), and whether it was concealed from participants and trial personnel until recruitment was confirmed. Please also indicate whether any blocking was used, and if so, what were the block sizes?Response: Our randomization was stratified by weight status and concealed for all participants and the authors until recruitment was confirmed. We did not use blocking. We insert detailed information (pg line)
  11. Page 2 line 85: The trial was clearly registered retrospectively, as the date of registration was 10 January 2022. Please state that the trial was registered retrospectively to avoid doubt for the reader. Response: Yes, it was retrospective. the information was inserted (pg line)
  12. Page 2 line 90: please provide information about the reliability and validity of the questionnaire for the children's daily routine. Since the reference is not an English language resource many readers will not be able to access it to obtain this information themselves. Furthermore, please consider providing information about how many questions it has and how it is scored. Response: Detailed information about the questionnaire and references to previous use with Brazilian children are provided. We provided the results of the validity analysis that we conducted in the present sample (internal consistency and test-retest temporal stability).
  13. Page 3 2.3 Motor Skill Intervention Design and Implementation: A great amount of detail is provided for the MMC group. Please write another section providing the same level of detail about the control group. I recognize that there is some information provided in figure 1 however it is important for consistency to write the information in text also. Response: We agree. A detailed description of this group is provided now. 
  14. Page 3 line 132: remove 'were' in "Children were participated in..." Response: we correct the sentence.
  15. Page 3 line 133-134: please explain how the healthy eating habits sessions were structured in as much detail as possible (contents, framework etc.) and also explain how the healthy eating habits sessions were positioned in the 28 weeks (e.g., all motor skills sessions weeks 1-21 then all healthy habits sessions weeks 21-28 or interspersed?). If it is your intention to report the healthy eating habits results/intervention in another manuscript, please state your intention for this paper that you will focus only on the motor skills portion of the intervention. Response: We provided further explanation about the eating habits session. Unfortunately, we do not measure changes in children's eating habits. 
  16. Page 5 Figure 1: Clarify at which point the assessment pre-intervention took place: before or after randomization? In the figure it appears that it happened after however I understand that TGMD-2 was applied first before randomization to eliminate participants who did not meet the motor delay criteria. Response: We changed the figure since the TGMD-2 and BMI were conducted before the randomization.
  17. Page 5 Figure 1: Break down the number of participants lost to follow up per treatment group rather than as a whole Response: the changes were made.
  18. Page 5 Data Analysis: since your trial was registered retrospectively, readers will not be able to know whether your analyses were prospectively designed. Consider attaching a statistical analysis plan (dated) as a supplementary file to increase the quality of your reporting. You would have had to submit this as part of your ethically approved protocol. Response: Although the register for RCT was done retrospectively in the national database, it was done prospectively in the university committee. The statistical planning procedures were conducted prospectively and the sample size estimation. We recognized that it was unclear in the methods section; we rewrite for clarity; we also provided supplementary tables with all the means, SD, and statistical procedures. 
  19. Page 5 Results: for transparency please provide the raw means and standard deviation scores and mean differences pre- and post-intervention for ALL outcomes that were analysed using ANOVA, even if it has to be in a supplementary data file. You have shown the graphs however these are difficult to extract data from, and it is likely that other researchers will seek this data for purposes of meta-analysis. Furthermore it is important to enable your statistical analyses to be verified. You may also consider publishing a copy of your dataset in a data repository which would serve the same purpose. Response: We now provided several supplementary tables with means, SD, means differences pre- to post-test (delta), and all the statistical results, including the delta comparisons.
  20. Page 5 line 186: please use a consistent acronym throughout the whole paper for the intervention group, ideally MMC so that it does not also contain CG and be confused with the control group. Response: We revised all acronyms, and it is consistent now across the paper. We used the MMC for the Mastery Motivational Climate and MCG for the Mastery Climate Group.
  21. Table 1: I am not sure why there are four columns. Is this because it is broken up into O/O and AW children? If so, there is a missing row at the top of the table that shows which columns are O/O and which columns are AW. Response: sorry, the first line was missing regarding the MCG and CG. We inserted it now. 
  22. Table 1 legend: should it read p<0.05 not p<0.5? Response: We correct the mistake.
  23. Figures: This may have been done by the typesetter, but the graphs in figures are extremely small and are impossible to read. If it is within your control, please provide graphs that are as large as possible and in a high quality image format. Response: We enlarged the source and provided now figures with high quality. 
  24. Page 11: Consider adding a separate section where you detail some limitations of the study, for example that: 1. physical activity behaviours were measured subjectively using a self-report measure which may introduce bias (future studies should consider objective measurement with accelerometers), 2. outcomes were measured immediately post-intervention, so the longevity of changes in behaviour are not known. Given the environment (low-socioeconomic, low-resource) and because children have limited formal physical activity opportunities, it is critical to understand whether this intervention leads to a long-term change or whether ongoing intervention is required to sustain decreases in BMI, increased PA etc. Response: We appreciate the suggestion. A new limitation section was inserted. 
  25. Page 12 line 404: please expand the acronyms for the trial funders if relevant. Response: we expanded the acronyms. 

Reviewer 2 Report

The aim of the paper was to present outcomes of RCT study on effects of a specific form of intervention aimed at increasing physical activity of children. Its main contribution is that it a step towards an assessment of the efficacy of the intervention based on particular methodology (Mastery Motivational Climate).

Both in terms of clarity and appraisal of its scientific soundness, the paper would benefit substantially if it included a more in-depth discussion of the activities proposed to the control group. They have been concisely described in Figure 1, but I would suggest adding a description within the main text as well. Furthermore, an explanation why this particular set of games and activities was chosen as a 'placebo' would help with interpretation of the interventions' effects identified. It also not explained convincingly enough why at least some (e.g., conflicts, distractions, changing tasks) measurement of time was impossible for the control group.

Furthermore, I also think that the paper would benefit from including more details on the schedulling and spatial organistation of the activities. For the former - an important, in terms of possible interpretation, information would be the timing of the parents participation. I have also missed an information about timing of the monitoring period - when did take place? For the five days directly after end of the intervention? This information would be important for the appraisal of the sustainability of the effects observed. As for the latter matter (spatial organisation), the most important question would be whether was it comparable for the control group?

It is not entirely clear for me, what do you mean by the 'ANOVA with repeated measures on the time factor'. The description and Figure 1 suggest that the continous outcome variables were measured only twice (before and after intervention). Also, the intervention is of a binary nature. I would be grateful for an explanation in this respect. As a minor comment regarding description of the data analysis - I think that the term 'time main effects' might be misleading here. I presume it concerns the time allocation to different activities during intervention (e.g., distractions, conflicts, free play?)? I would also recommend considering explicitly stating in this part what are the dimensions (MCG vs CG, O/O vs AW) analysed within the interaction effects in particular setups. It can be inferred from the results section and the research design itself, but I think the text would be easier to follow for a relatively general audience with this information clearly stated.

The description of parents' involvement and nutritional content in lines 154-157 is not entirely clear to me - did the parents participate in the activities, or they stood by during the sessions during when the children practiced? Moreover, was the nutritional content used during parents-attented sessions? Was it specifically aimed at parents or children? Did all the parents took part? This particularly important since some of the effects observed might be a result of this part of the intervention, rather then participation of children themselves. While it seems impossible to disentagle the contribution of the two parts of the intervention within the conducted research, perhaps it would be possible to comment on the issue more extensively within the discussion (lines 326-333). In particular, is the nutritional education an inherent part of the MMC? If not, what was the motivation behind its use within the programme? Does the pre-existing literature allow to discuss the potentially confounding effect of the 'nutritional part' of the intervention? In the context of the impact on parents, is it possible that children from the same household were involved in the programme (in particular, both intervention and control group)?

The construction and description of table 1 is unclear, as much as it contains a lot of information indeed. Nevertheless, I feel a reader could be further assisted in attempts to understand it. In particular, only based on the description in the text I have inferred that the first two columns refer to the O/O group and two other to the AW group. Furthermore, a note states that data on four children was missing, whereas Figure 1 states that 22 children ceased participation. Does this discrepancy comes from the moment of the cease in participation? I would recommend explaining it with a greater precision. Furthermore, I have not found what does the '##' note refer to, nor what does it state (what does 'Only Child' mean in this context?). How was the Chi2 and Macnemar tests applied? (For example, for the same 2x2 'subtables' Pre and Post, for each activity separately? Based on the description and standard use, this seems to be the case for the Macnemar test, but it is not clear for the former test). Alternatively, the tests (especially the Chi2 test) could be explained more extensively in line 173. Lastly, is the 'p' mentioned in the note a critical p-value (if so, is it really .50? Or, rather .05?) or parameter of the binominal parameter underlying the exact test (in this case, I would expect an equality, p = 0.5)?

Furthmore, as for Table 1, the first group (as I understand, MCG O/O) seems to differ substantially from the other three in terms of using bus, though the sample sizes are small. Difference in comparison to CG O/O would be particularly important. Have you tested it formally for the statistical significance (for example, was the Chi2 test used for this)? Another substiantial difference can be observed in case of running.

I would consider adding a (very) short information (aimed at a relatively general audience) on the achievement theory (in general) and the TARGET structure themselves in the introduction (their background and relation of the former to the MMC).

On a more general note, in terms of conclusions (and statement in lines 280-281 regarding the promise), I would recommend considering extending the discussion to include the matter of sustainability of the observed effects. The previously mentioned matter of the timing of the after-intervention measurement is of importance in this respect. For example, would you consider the effects as likely to be sustained for a longer time if the intervention was one-time only? Would a sustained programme have sustained effect or, rather, they could possibly be partially subject to a novelty effect? Obviously, these questions are beyond the scope of the research presented in the paper, but perhaps it would be possible to refer to them within the discussion? Is it possible to infer in this matter based on pre-existing literature? Or, rather, it would be worthwhile to suggest them as a potential for future research?

Within the conclusions, a statistically significant decrease in waist circumference is mentioned (line 331), but it is not presented within the results section (lines 228-234).

Furthmore, was the pre-intervention difference between O/O and AW mentioned in line 359 children in terms of engagement in distractions formally tested? It seems to me that it was not reported in subsection 3.5. Within the subsection 4.5 would also explicitly state that for this part of outcomes the interpretations should be drawn with greater caution due to an effective lack of a control group in this regard.

As for practical implications (especially in the policy context), would it be possible to refer to or comment on the relative costs of a MMC-based intervention (in comparison to a 'standard' one?), even if only qualitatively? How much more effort/resources would it take to scale up such a programme?

Technical/specific/minor comments:

  • does 'playtime' involve both physical activities and any other activies in free time, as long as use of TV and computers is excluded?
  • lines 90-95 - were the questionnaires filled in by parents and the activies were self-reported (by parents)? Or by the researchers?
  • line 97 - I would consider explaining (shortly) the term 'global self-worth'
  • line 47 - wouldn't 'Furthermore' be a better word instead of 'Nevertheless'?
  • lines 64-70 contain description of a typical content of the 'Materials and Methods' instead of the actual introduction to the part;
  • line 285 - wouldn't 'recommendation' be a more appropriate word than 'endorsement'?
  • lines 263, 295, 297, 306, 311, 316, 318, 357, 370, 376: MC instead of MMC was used
  • line 290 - wouldn't 'active' be a more appropriate word than 'dynamic'?
  • line 325 - wouldn't 'promising' be a more appropriate term than 'exciting'? As well as 'global trend' or 'global context' instead of 'global scenery'?
  • line 363 - wouldn't 'diversified' or 'varied' be a more appropriate term than 'full of possibilities'?
  • line 388 - wouldn't '...was an assesment of an MC intervention' be a more appropriate statement than '...was an MC intervention'?
  • line 390 - wouldn't 'assess' be a more appropriate term than 'promote'?
  • line 395 - wouldn't 'common perception' be a more appropriate term than 'common sense'?

    Best regards

Author Response

Thank you for the nice and constructive comments, all your suggestions were addressed.

1. The aim of the paper was to present outcomes of RCT study on effects of a specific form of intervention aimed at increasing physical activity of children. Its main contribution is that it a step towards an assessment of the efficacy of the intervention based on particular methodology (Mastery Motivational Climate). Both in terms of clarity and appraisal of its scientific soundness, the paper would benefit substantially if it included a more in-depth discussion of the activities proposed to the control group. They have been concisely described in Figure 1, but I would suggest adding a description within the main text as well. Response: We agree and inserted information about the control group. 

2. Furthermore, an explanation why this particular set of games and activities was chosen as a 'placebo' would help with interpretation of the interventions' effects identified. It also not explained convincingly enough why at least some (e.g., conflicts, distractions, changing tasks) measurement of time was impossible for the control group. Furthermore, I also think that the paper would benefit from including more details on the schedulling and spatial organistation of the activities. Response: We provided further information about the control group activities. Regarding the children engaged in the control group, our first choice during the RTC design was better to understand the children engagement in mastery climate intervention, considering the lack of studies addressing this issue; more traditional approaches to teach, similar to the one conducted in the control group, have more robust studies regarding engagement. We strongly regret that and therefore emphasize that it was a limitation in the present study. 

3. For the former - an important, in terms of possible interpretation, information would be the timing of the parents participation. I have also missed an information about timing of the monitoring period - when did take place? For the five days directly after end of the intervention? This information would be important for the appraisal of the sustainability of the effects observed. As for the latter matter (spatial organisation), the most important question would be whether was it comparable for the control group? Response: We inserted the information about the time we monitored children's activity at home (first and last week of intervention). Parents participated in six lessons; we also inserted more detailed information about this issue.

4. It is not entirely clear for me, what do you mean by the 'ANOVA with repeated measures on the time factor'. Response: We rewrite the data analysis section for clarity, we conducted a 2 (groups: MCG and CG) x 2 (nutritional status: O/O and AW) x 2 (time: Pre- to Post-intervention) ANOVA with a repeated measure on the last factor (time)

5. The description and Figure 1 suggest that the continous outcome variables were measured only twice (before and after intervention). Also, the intervention is of a binary nature. I would be grateful for an explanation in this respect. Response: We understand the importance of having more time points outcome measures during the intervention; however, conducting an intervention is very time demanding, and it is hard to include other time points measured along with the intervention. We want to point out that we addressed several outcomes, advancing previous research, but we will keep this suggestion for the following studies. 

6. As a minor comment regarding description of the data analysis - I think that the term' time main effects' might be misleading here. I presume it concerns the time allocation to different activities during intervention (e.g., distractions, conflicts, free play?)? I would also recommend considering explicitly stating in this part what are the dimensions (MCG vs CG, O/O vs AW) analysed within the interaction effects in particular setups. It can be inferred from the results section and the research design itself, but I think the text would be easier to follow for a relatively general audience with this information clearly stated. Response: sorry for the confusion; we are referring to the pre-to-post intervention effect. We agree that it was not clear and rewrite the data analysis section. 

7. The description of parents' involvement and nutritional content in lines 154-157 is not entirely clear to me - did the parents participate in the activities, or they stood by during the sessions during when the children practiced? Moreover, was the nutritional content used during parents-attented sessions? Was it specifically aimed at parents or children? Did all the parents took part? This particularly important since some of the effects observed might be a result of this part of the intervention, rather then participation of children themselves. While it seems impossible to disentagle the contribution of the two parts of the intervention within the conducted research, perhaps it would be possible to comment on the issue more extensively within the discussion (lines 326-333). Response: we provided further explanation about parental participation to clarify this issue. We also extended our discussion and limitation sections.

8. In particular, is the nutritional education an inherent part of the MMC? If not, what was the motivation behind its use within the programme? Does the pre-existing literature allow to discuss the potentially confounding effect of the 'nutritional part' of the intervention? Response: In fact, nutritional education is a novelty implemented in the present MMC intervention. The rationality behind our design was a large number of children with motor delays and obesity/overweight in previous studies. 

9. In the context of the impact on parents, is it possible that children from the same household were involved in the programme (in particular, both intervention and control group)? Response: No, none of the children.

10. The construction and description of table 1 is unclear, as much as it contains a lot of information indeed. Nevertheless, I feel a reader could be further assisted in attempts to understand it. In particular, only based on the description in the text I have inferred that the first two columns refer to the O/O group and two other to the AW group. Response: Sorry for the confusion. the first line of the table was missing, maybe during the configuration on the templates. We correct the mistake.

11. Furthermore, a note states that data on four children was missing, whereas Figure 1 states that 22 children ceased participation. Does this discrepancy comes from the moment of the cease in participation? I would recommend explaining it with a greater precision. Response: We clarify that four families fail to return the questionnaires in the table. Children participated in the study but four families did not return the routine questionnaire.

12. Furthermore, I have not found what does the '##' note refer to, nor what does it state (what does 'Only Child' mean in this context?). Response: The information was not relevant for this study. We deleted the information. 

13. How was the Chi2 and Macnemar tests applied? (For example, for the same 2x2' subtables' Pre and Post, for each activity separately? Based on the description and standard use, this seems to be the case for the Macnemar test, but it is not clear for the former test). Alternatively, the tests (especially the Chi2 test) could be explained more extensively in line 173. Lastly, is the 'p' mentioned in the note a critical p-value (if so, is it really .50? Or, rather .05?) or parameter of the binominal parameter underlying the exact test (in this case, I would expect an equality, p = 0.5)? Response: We used the Chi-squared to compare the groups in the categorical variable and the Macnemar to compare the pre-to-post intervention. We inserted further information to clarify this issue. 

14. Furthmore, as for Table 1, the first group (as I understand, MCG O/O) seems to differ substantially from the other three in terms of using bus, though the sample sizes are small. Difference in comparison to CG O/O would be particularly important. Have you tested it formally for the statistical significance (for example, was the Chi2 test used for this)? Another substantial difference can be observed in case of running. Response: We did not find any differences between groups in the daily routine using the Chi2, maybe due to the small sample size. The only significant changes were related to pre-to-post tests. 

15. I would consider adding a (very) short information (aimed at a relatively general audience) on the achievement theory (in general) and the TARGET structure themselves in the introduction (their background and relation of the former to the MMC). Response: We inserted a new paragraph in the introduction.

16. On a more general note, in terms of conclusions (and statement in lines 280-281 regarding the promise), I would recommend considering extending the discussion to include the matter of sustainability of the observed effects. The previously mentioned matter of the timing of the after-intervention measurement is of importance in this respect. For example, would you consider the effects as likely to be sustained for a longer time if the intervention was one-time only? Would a sustained programme have sustained effect or, rather, they could possibly be partially subject to a novelty effect? Obviously, these questions are beyond the scope of the research presented in the paper, but perhaps it would be possible to refer to them within the discussion? Is it possible to infer in this matter based on pre-existing literature? Or, rather, it would be worthwhile to suggest them as a potential for future research? Response: We agree, and further information and recommendation were inserted in the discussion section. 

17. Within the conclusions, a statistically significant decrease in waist circumference is mentioned (line 331), but it is not presented within the results section (lines 228-234). Response: we addressed the significant decrease in the MCG of children with O/O in. the results section.

18. Furthmore, was the pre-intervention difference between O/O and AW mentioned in line 359 children in terms of engagement in distractions formally tested? It seems to me that it was not reported in subsection 3.5. Within the subsection 4.5 would also explicitly state that for this part of outcomes the interpretations should be drawn with greater caution due to an effective lack of a control group in this regard. Response: We did not compare children with O/O with children with AW, only the changes over time. We agree about the caution regarding engagement results; we clearly state this issue in the paper now. 

19. As for practical implications (especially in the policy context), would it be possible to refer to or comment on the relative costs of a MMC-based intervention (in comparison to a 'standard' one?), even if only qualitatively? How much more effort/resources would it take to scale up such a programme Response: we insert a sentence about this issue 

Technical/specific/minor comments:

  • does 'playtime' involve both physical activities and any other activies in free time, as long as use of TV and computers is excluded? Response: yes, we provided further explanation 
  • lines 90-95 - were the questionnaires filled in by parents and the activities were self-reported (by parents)? Or by the researchers? Response: the parents filed the daily routine questionnaire. We provided further details n. the methods section. 
  • line 97 - I would consider explaining (shortly) the term' global self-worth' Response: yes, the Harter definition of this dimension is inserted now in the paper 
  • line 47 - wouldn't 'Furthermore' be a better word instead of 'Nevertheless'?Response: thanks for the language advice, "furthermore" is used now. 
  • lines 64-70 contain description of a typical content of the 'Materials and Methods' instead of the actual introduction to the part; Response: Sorry for the mistake; we deleted the information.
  • line 285 - wouldn't 'recommendation' be a more appropriate word than 'endorsement'? Response: We agree.
  • lines 263, 295, 297, 306, 311, 316, 318, 357, 370, 376: MC instead of MMC was used Response: we correct the mistake along with the paper
  • line 290 - wouldn't 'active' be a more appropriate word than 'dynamic'? Response: we agree.
  • line 325 - wouldn't 'promising' be a more appropriate term than 'exciting'? As well as 'global trend' or 'global context' instead of 'global scenery'?Response: we agree.
  • line 363 - wouldn't 'diversified' or 'varied' be a more appropriate term than 'full of possibilities'? Response: We agree
  • line 388 - wouldn't '...was an assesment of an MC intervention' be a more appropriate statement than '...was an MC intervention'? Response: we rewrite the sentence. 
  • line 390 - wouldn't 'assess' be a more appropriate term than 'promote'?Response: we switch the word for "implementing."
  • line 395 - wouldn't 'common perception' be a more appropriate term than 'common sense'? Response: we agree